Review

 

**Subject Area:**
cellular biology/biochemistry/genomics

phase separation, chromatin organization, nuclear bodies, cellular ageing, transcription

**Author for correspondence:**
Argyris Papantonis
e-mail: argyris.papantonis@med.uni-goettingen.de

# Modes of phase separation affecting chromatin regulation

Spiros Palikyras and Argyris Papantonis

Institute of Pathology, University Medical Center, Georg-August University of Göttingen, Robert-Koch-Str. 40, 37075 Göttingen, Germany

AP, 0000-0001-7551-1073

It has become evident that chromatin in cell nuclei is organized at multiple scales. Significant effort has been devoted to understanding the connection between the nuclear environment and the diverse biological processes taking place therein. A fundamental question is how cells manage to orchestrate these reactions, both spatially and temporally. Recent insights into phase-separated membraneless organelles may be the key for answering this. Of the two models that have been proposed for phase-separated entities, one largely depends on chromatin–protein interactions and the other on multivalent protein–protein and/or protein–RNA ones. Each has its own characteristics, but both would be able to, at least in part, explain chromatin and transcriptional organization. Here, we attempt to give an overview of these two models and their studied examples to date, before discussing the forces that could govern phase separation and prevent it from arising unrestrainedly.

## 1. Nuclear sub-compartmentalization via phase separation?

The core concept of phase separation itself is not really new. Already in 1899, the American biologist E. B. Wilson had observed that after squishing starfish eggs, the spherical formations in the cellular goo were able to fuse with each other, but only if they were of the same type [1]. Nowadays, it is known that similar droplets also exist in eukaryotic nuclei, though variable in their sizes, abundance and properties. Given their dynamic properties, and the fact that nuclear compartmentalization cannot be static in order to accommodate and coordinate the huge variety of biochemical reactions that take place therein, a major question arising is: how might such phase-separated nuclear entities contribute to the organization and regulation of chromatin? In the light of recent data on phase separation-driven compartmentalization, this review aims at providing some insight on the key characteristics of nuclear phase-separated formations, on how phase separation may regulate chromatin organization and on the forces that restrain phase separation from occurring in a non-orchestrated manner.

The development of technologies like whole-genome chromosome conformation capture (Hi-C) allowed for a reappraisal of chromatin organization [2]. As a result of numerous Hi-C studies, we now understand that chromosomes are generally divided into alternating Mbp-long compartments: the A- (mostly transcriptionally active) and the B- (mostly transcriptionally inactive) compartment. At the sub-Mbp scale, these A-/B-compartments further consist of 'loop domains' stabilized by the chromatin-bound insulator CTCF and the cohesin ring protein complex [3,4], and/or of 'topologically associating domains' (TADs) harbouring stretches of chromatin that tend to physically interact with one another more frequently than with chromatin in other TADs [5–7]. Despite the fact that the nucleus appears well compartmentalized in Hi-C data, this compartmentalization must be dynamically orchestrated and amenable to acute regulation.

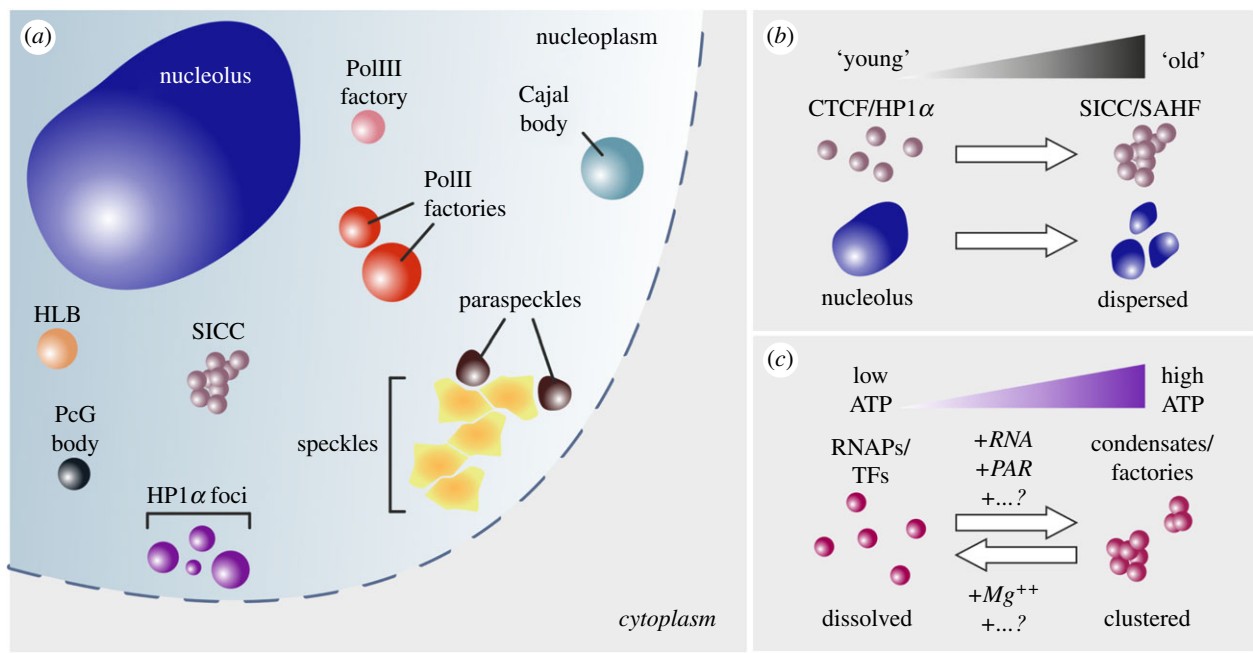

**Figure 1.** Phase separation in the cell nucleus. (*a*) Cartoon depicting different kinds of membraneless entities in mammalian cell nuclei, ranging from the large nucleolus (blue; 0.2–3.5 μm) to transcription factories (red/pink, including histone locus bodies, 'HLB', orange; approx. 0.1 μm), Cajal (green; 0.3–1.0 μm) and Polycomb bodies (black; 0.2–1.5 μm) or splicing speckles (yellow; 20–50 nm) and paraspeckles (brown; 0.2–1.0 μm). (*b*) Nuclear phase-separated entities such as SICCs or SAHFs, forming on the basis of HP1α (purple in (*a*); less than 0.5 μm) or CTCF (light purple in (*a*); 0.5–1.5 μm), become most evident under conditions of cellular ageing. At the same time, the nucleolus changes in shape and dispersion in chronologically aged or longevity-related conditions. (*c*) Persistence of elevated nuclear ATP levels, in conjunction with chromatin/protein modifications and high local RNA titres, aid in the maintenance of supramolecular condensates (factories) by TFs and the general transcription machinery, while low ATP levels, $Mg^{++}$ cations and additional insofar unknown factors will deter and/or reverse such phase separation in the nucleoplasm.

Along these lines, different studies have now proposed that phase separation might, at least in part, control transcription [8–10] and, as a result, genome architecture and accessibility [11,12] via the formation of a large variety of membraneless nuclear bodies (figure 1*a*). Two conceptually different mechanisms have been proposed to explain how this might be achieved. On one hand, 'polymer–polymer phase separation' (PPPS) can promote the assembly of chromatin globules in the nucleus via proteins which interlink its different segments; on the other, 'liquid–liquid phase separation' (LLPS) can lead to droplet formation in the cytoplasm and nucleoplasm, stabilized by multivalent interactions among the participating components [13]. In other words, in the case of PPPS, molecules need not actively bind to one another but are dependent on the availability of chromatin (and most probably of other contributing factors), while in LLPS, bridging interactions with nucleic acids are not a prerequisite for droplet formation compared to the interactions between disordered domains of the contributing proteins (for a comparison of the two, see table 1).

## 1.1. PPPS in chromatin organization

Chromatin in cell nuclei is by and large flexible and accessible, due to the ability of nucleosomes to locally fluctuate [35,36], hence the 10 nm chromatin fibre [37] acquires a more 'liquid-like' (rather than static) behaviour [38]. In this 'liquid-like' state, nucleosomes found in close proximity can induce PPPS with the contribution of certain bridging interactions. Such interactions may, for example, occur among histone tails (and bound factors thereon). In a decreasing order of magnitude, histone H4 tails seem to participate in interchromosomal interactions, accompanied by H3 and H2A/B tails, although the latter seem to mainly help maintain fibre-to-fibre interactions [14]. Cohesin and condensin have also been identified as major components in diverse processes of genome folding [39,40]. The cohesin complex (SMC1A, WAPL and NIPBL) binds to chromatin and mediates its compaction and looping presumably by 'loop extrusion' [15,41,42]. The CCCTC-binding factor (CTCF) also critically participates in this process, as it almost invariably co-localizes with cohesin at TAD boundaries as well as at the CTCF loop anchors [7,15,16]. Interestingly enough, there exist strong indications that CTCFs phase separate upon entry of human primary cells into senescence, a state of irreversible cell cycle arrest. It appears that these 'senescence-induced CTCF clusters' (SICCs; figure 1*a,b*) constitute an intermediate between PPPS and LLPS, as they remain bound to chromatin while large multimeric SICCs are created on top [18]. In support of this potential CTCF mode of clustering, come data of RNA-mediated CTCF interactions that also affect the spatial organization of chromatin in mouse ES cells at the sub-Mbp level [17]. Of course, many other proteins carrying DNA-binding motifs alongside disordered domains exist in mammalian cells and could in theory bridge chromatin and drive PPPS [43], thus affecting the spatial organization of different genomic compartments. Recent studies [19,20] describe such a role for HP1α, which marks heterochromatic regions throughout the genome and, through association with the histone methyltransferase SUV39H1, can spread along the DNA fibre. HP1α uses its N-terminal chromodomain to interact with H3K9-methylated nucleosomes, while self-interacting to other HP1α molecules via its C-terminal 'chromo-shadow' domain [30]. Their clustering, on the basis of the PPPS model, does not primarily rely on direct interactions between the participating bridging

royalsocietypublishing.org/journal/rsob Open Biol. 9: 190167

**Table 1.** Major features and components of PPPS and LLPS (relevant references in square brackets).

| | polymer–polymer phase separation | liquid–liquid phase separation |
|---|---|---|
| structure | chromatin-associated proteins cross-linking different chromatin fragments | chromatin-associated proteins developing multivalent interactions with each other |
| chromatin dependence | high; on the number/density of chromatin binding sites | low; droplets lacking a chromatin scaffold can also be stable |
| interaction dependence | low need of interactions among bridging proteins | abundance of protein interactions within the droplet |
| fluid microenvironment | same composition within and outside of the compartment | different composition inside and outside of the droplet |
| implicated proteins and structures | histone-tail modifications [14]; cohesin complexes [7,15–17]; CTCF [18]; HP1 proteins [19,20] | P-bodies [21]; stress granules [22]; nucleoli [23]; paraspeckles [24,25]; Cajal bodies, PML bodies, and transcription factories [25–29] |
| relevant biological processes | senescence-induced CTCF clustering [18]; heterochromatic spreading [30,31]; transcriptional regulation [8–10,32,33] | amyloid formations in Alzheimer's, Parkinson's synuclein plaques and ALS plaques [34] |
| identification assays | 3C-based techniques super-resolution imaging | FRAP analysis ultrafast-scanning FCS protein engineering |

factors [31], and each newly established globule contains the same nucleoplasmic fluid as its microenvironment. Thus, the human HP1$\alpha$ protein can play, via phase separation, a central role in B-compartment formation [44]. However, it is the fraction of chromatin each such globule occupies and the exact factors bound to that chromatin stretch that determines the final properties and extent of clustering [13]. Interestingly, and perhaps similar to senescence-induced clustering of CTCFs, gradually more intense HP1$\alpha$ foci appear in cell nuclei as cells enter replicative [17] or, more strikingly, oncogene-induced senescence, called 'senescence-associated heterochromatic foci' (SAHFs; figure 1a,b) [45,46]. If one now also considers how chronologically old nuclei display perturbed nucleolar formations (figure 1b), and that eukaryotic longevity correlates well with multiple nucleoli of small sizes [47], it is attractive to speculate that cellular ageing is also related to regulated phase separation.

## 1.2. LLPS driving nuclear droplet formation

Cells harbour organelles in their nucleoplasm (and cytoplasm) that can form and separate from their microenvironment in the absence of a membrane enclosure [34,48]. Such membraneless organelles regularly acquire liquid-like properties like the ability to fuse, to maintain different consistencies inside and outside the droplet, and to rapidly exchange components with their surroundings [13,21]. The formation of these phase-separated droplets is maintained mostly via multivalent interactions between the low complexity intrinsically disordered regions (IDRs) of the proteins participating in the assembly [49,50]. These low-complexity domains are over-populated by particular amino acid residues, in arrays of only a few different residues to long stretches containing just a single amino acid; this allows the respective proteins to assume multiple conformations and, thus, to not necessarily reproduce the same secondary structure every time [51,52]. Recently, evidence was presented of many cases where these interactions are stabilized with the assistance of RNA molecules [53–56]. In these RNA–protein droplets in the cytoplasm (e.g. P-bodies [21], stress granules [22]) or in the nucleus (with the prototypic example of the nucleolus [23]),

RNA might act as a regulatory element controlling their size and constitution, as recently reported [57].

The prominent liquid phase-separated nucleolus has been extensively studied [23,58–60], primarily acts to produce the ribosomal subunits and is made up of a variety of proteins and RNA. A later study suggests that due to its phase separation abilities, the nucleolus could act as a protein quality control compartment inside the nucleus, especially under stress conditions [61]. Nucleoli muster many characteristics of phase-separated droplets, such as rapid signal recovery following FRAP (fluorescence recovery after photobleaching) analysis, fusion of smaller sized droplets into a larger droplet-like conformation, and extensive exchange of molecules between the two sides of the separated phase [62]. Given that nucleolar organizing regions (NORs) from different chromosomes come together in three-dimensional space to form nucleoli, it is evident that such structural changes of this large organelle will invariably impact the relative positioning and folding of mammalian chromosomes.

Along the same lines, multiple nuclear membraneless formations have been identified that exhibit such characteristics. For example, paraspeckles are discrete bodies found in nuclei and created on the basis of protein–protein and protein–RNA interactions [24,25]. Their assembly is highly dynamic as they become apparent in human cells only upon differentiation [24], and DNA is typically absent from the interior of these liquid-like droplets, while at the same time, there exists evidence of lnRNAs being used as scaffolds for their formation and maintenance [25,63,64]. Similarly, Cajal bodies (CBs), histone locus bodies (HLBs) and promyelocytic leukaemia (PML) bodies are all formations that have been shown to form phase separated-like droplets in the nucleoplasm (figure 1a) and have the ability to accumulate multifarious macromolecules from their surrounding interchromatin regions [25–29]. However, for all the above-mentioned droplets a certain thermodynamic threshold has to be reached in order for them to form, and once these LLPS bodies are large enough, they can expand without a need for nucleation sites [11,13]. Obviously, the association of all of these bodies with chromatin makes them at the same time important for its overall organization in three-dimensional nuclear space.

## 2. Phase separation and transcriptional regulation

The creation of liquid condensates in nuclei, and the exclusion of chromatin from many of them after acquiring a certain size, may markedly restructure the nuclear environment. The energy stored in the 'chromatin matrix' during this restructuring directly affects the size and distribution of these droplets [65], which is favoured in lower-density chromatin regions [11]. This set of preferences in positioning and compartmentalization may supervise reorganization of the genome in response to stimuli and, as a result, gene expression itself.

The regulation of gene expression is based on the ability of the transcription machinery to assemble at specific genomic loci. This is administered by transcription factors (TFs) which, via their DNA-binding domains (DBDs) and activation domains (ADs), bind specific positions at enhancers and promoters. While DBDs are well structured, interestingly, the ADs of many TFs contain IDRs [66], similar to the ones involved in the establishment of phase-separated droplets. A variety of TFs have been shown to interact with similar groups of coactivator complexes [67,68] and it is proposed that these interactions are mainly maintained via the ADs. For instance, the AD of GCN4 was shown to interact with the Med15 subunit of the Mediator complex, in a 'fuzzy protein–protein complex' [69]. The large multi-component Mediator complex interacts with various TFs and the RNA polymerase II, apparently creating phase-separated condensates to promote gene activation [32,70,71]. Such condensates, already discernible using light microscopy, are particularly strong when involving stretches of multiple strong enhancers, known as 'super-enhancers', and phase separation can explain their engagement with cognate gene promoters and the concomitant transcriptional activation [9,72]. This model offers the advantage, at least for the loci associated with 'super-enhancers', that gene expression control becomes less stochastic and less 'noisy' [73–76], while providing a framework able to explain the synchronous activation of genes regulated by the same set of enhancers [77]. Notably, though, such a stochastic yet biophysically tuneable high local concentration of relevant TFs and RNA polymerases aligns well with the 'transcription factory' model, whereby transcription occurs focally at discrete nucleoplasmic sites for gene loci transcribed by either RNA polymerase II or III [78,79]. Thus, bridging the two concepts now allows us to use phase separation as the underlying mechanism that explains the acute, reversible and tuneable formation of transcription factories, thereby directly impacting the 'bursting', noise and tuneability features of gene transcription itself. Moreover, these merged concepts and mechanisms also put the relative positioning of enhancers, promoters, silencers and insulators into play to explain how the activation, repression or insulation of different spatial neighbourhoods (from individual loops to compartments) is indeed dynamic, highly interconnected and, critically, tuneable in response to the extracellular cues and challenges a cell faces through its life cycle [80]. But how is such fine-tuning achieved?

## 3. Some mechanisms controlling phase separation

The nuclear environment is crowded, considering the many thousands of macromolecules cohabitating a space of just a few $\mu m^3$. This then raises the following questions. In which way do cells orchestrate a phenomenon such as phase separation and prevent it from occurring uncontrollably? And how can this be rendered reversible? Recent studies suggest that ATP may play a central role in the regulation of liquid-like condensate formation [12,60]. The proteins and RNA that participate in phase-separated droplets are inherently able to form aggregates, typically dependent on their local concentration, the microenvironment and the supplies of energy. Apart from being the 'energy currency' of cells, ATP has also been attributed a role as a hydrotrope able to destabilize protein aggregates [81]. For example, nucleolar viscosity is a partially ATP-dependent condition [23], and any given droplet can sustain a more liquid-like interior [82]. Another observation in support of such a role for nuclear ATP was that following hormone-induced chromatin reorganization, ATP levels were maintained notably high for much longer (approx. 30 min) than compared to the changes in chromatin kinetics in response to the stimulus (occurring in a 1–15 min window) [12,83]. Why does ATP persist in nuclei? According to this hydrotrope model, mM ATP levels and, at the same time, markedly lower $\mu M$–nM $K_m$ of cellular ATPases can conceivably be used to maintain a liquid-like state in the nucleus via actively preventing aggregation of its components and keeping this microenvironment out of equilibrium [50] (figure 1c).

At the biochemical level, interactions within droplets are mostly retained through weak and predominately hydrophobic interactions, but also through protein–protein and/or protein–nucleic acid interactions of electrostatic nature among residues in IDRs [12]. Post-translational modifications, like methylation, acetylation or PARylation, have also been reported to actively participate in the formation and/or disruption of phase-separated organelles by reinforcing or destabilizing these interactions [33,84–88] (figure 1c). Each of these mechanisms will only contribute to a particular degree to the assembly or disassembly of droplets, and further studies are required to understand their individual impact, especially on shaping chromatin.

## 4. Outlook

Phase separation does present an attractive model by which to explain nuclear compartmentalization and the regulation of the many diverse biochemical reactions taking place in cell nuclei. Despite the fact that most of the liquid condensates described above had been identified many years ago, it has only now become possible to mechanistically dissect their dynamics during different biological processes, ranging from the transcriptional to the translational level. In addition, although not covered in this review, membraneless organelles have been shown to have a key role in human pathophysiology [87,89–95]. Important emerging questions include the following. Why does only a particular (albeit larger than perhaps initially assumed) fraction of molecules have the ability to phase separate? Which are the signalling cues triggering such transitions? How may the nuclear environment regulate the generation of biomolecular condensates? These questions do not only address the role of phase separation in chromatin organization and regulation, but we believe that such basic knowledge on this phenomenon is bound to also shed light on how phase-separated nuclear entities arise and how they are modulated to exert control over rapid and precisely regulated nuclear processes. Still, since the phase separation field is still in its infancy,

royalsocietypublishing.org/journal/rsob    Open Biol. 9: 190167

the criteria and methodological approaches used to characterize the formation of phase-separated droplets and its outcomes must be constantly revisited and updated. Nonetheless, our perception of the cellular interior has been revolutionized, and this will surely allow a step forward in our efforts towards the decoding of the functional complexity of cellular processes.

Data accessibility. This article does not contain any additional data.

Competing interests. We declare we have no competing interests.

Funding. This work is supported by the German Ministry for Research (DFG) via funding of the TRR81 network (grant no. INST 160/697-1)

Acknowledgements. We wish to thank members of the Papantonis laboratory for discussions.

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
