## [Reviewer comments · Open Biology]

Review History

RSOB-19-0167.R0 (Original submission)

Review form: Reviewer 1

Recommendation

Accept with minor revision (please list in comments)

Do you have any ethical concerns with this paper?

No

Comments to the Author

In this review, Palikyras and Papantonis, discuss recent findings that support the role of phase separation in 3D chromatin organization and regulation of gene expression. They introduce the concept of phase separation, the origins, biological implications and potential mechanisms. They also describe two previously proposed models: polymer-polymer (PPPS) and liquid-liquid phase separation (LLPS). Overall, the subject is very timely and of high interest for a broader and specialized audience. The text is comprehensive and crisp, with few exceptions discussed below, and the figure is very well-organized. Below are some suggestions that I believe will further improve the manuscript:

-The discussion regarding the two phase separation models is not very clear for a non-specialized reader. Why for example phase separation of CTCF and HP1 belongs to the PPPS and not the LLPS category?

I think a table summarizing critical distinctions between PPPS and LLPS will be tremendously beneficial for any reader and will elevate the value and impact of this review. The table could include: key distinctive points regarding the organizational principles, known structures that belong in each category with references, proteins and other factors previously implicated in each model, in vitro and in vivo assays that allow assessment of PPPS or LLPS, and biological processes affected by PPPS and/or LLPS.

-The structure of the review is a bit confusing too. The authors start by providing a background on chromatin organization, then they switch to phase separation and then to gene regulation prior to describing mechanisms of phase separation. I think they should commit to the role of PS in chromatin organization from compartmentalization and TADs all the way to enhancer-promoter interactions. Alternatively, they could start by introducing PS (models, mechanisms) and then discuss in different chapters the implications in chromosome architecture and gene regulation.

- The title of the last chapter is a bit misleading, since this section really focuses on the role of ATP in phase separation.

-Interesting points are raised in the outlook, however many of the raised questions are not directly related to chromatin regulation but instead focus on general concepts related to phase separation.

Minor comments:

-Some sentences in the text seem overcomplicated and difficult to follow. For example: Lines 85-88 and lines 103-105.

-Lines 27-29 may lead to misinterpretation, since they distinguish loop domains from TADs. Although, indeed these terms describe different principles of organization, the resulting topological features are highly overlapping. Therefore, I suggest replacing the "and" in line 28 with "and/or".

Decision letter (RSOB-19-0167.R0)

27-Aug-2019

Dear Professor Papantonis

We are pleased to inform you that your manuscript RSOB-19-0167 entitled "Modes of phase separation affecting chromatin regulation" has been accepted by the Editor for publication in Open Biology. The reviewer has recommended publication, but also suggest some minor revisions to your manuscript. Therefore, we invite you to respond to their comments and revise your manuscript.

Please submit the revised version of your manuscript within 7 days. If you do not think you will be able to meet this date please let us know immediately and we can extend this deadline for you.

- 1) A text file of the manuscript (doc, txt, rtf or tex), including the references, tables (including captions) and figure captions. Please remove any tracked changes from the text before submission. PDF files are not an accepted format for the "Main Document".
- 2) A separate electronic file of each figure (tiff, EPS or print-quality PDF preferred). The format should be produced directly from original creation package, or original software format. Please note that PowerPoint files are not accepted.
- 3) Electronic supplementary material: this should be contained in a separate file from the main text and meet our ESM criteria (see <http://royalsocietypublishing.org/instructions-authors#question5>). All supplementary materials accompanying an accepted article will be treated as in their final form. They will be published alongside the paper on the journal website and posted on the online figshare repository. Files on figshare will be made available approximately one week before the accompanying article so that the supplementary material can be attributed a unique DOI.

Online supplementary material will also carry the title and description provided during submission, so please ensure these are accurate and informative. Note that the Royal Society will not edit or typeset supplementary material and it will be hosted as provided. Please ensure that the supplementary material includes the paper details (authors, title, journal name, article DOI). Your article DOI will be 10.1098/rsob.2016[last 4 digits of e.g. 10.1098/rsob.20160049].

- 4) A media summary: a short non-technical summary (up to 100 words) of the key findings/importance of your manuscript. Please try to write in simple English, avoid jargon, explain the importance of the topic, outline the main implications and describe why this topic is newsworthy.

Images

Data-Sharing

It is a condition of publication that data supporting your paper are made available. Data should be made available either in the electronic supplementary material or through an appropriate repository. Details of how to access data should be included in your paper. Please see <http://royalsocietypublishing.org/site/authors/policy.xhtml#question6> for more details.

Data accessibility section

Sincerely,

The Open Biology Team
 mailto:openbiology@royalsociety.org

Reviewer(s)' Comments to Author:

Referee:

Comments to the Author(s)

In this review, Palikyras and Papantonis, discuss recent findings that support the role of phase separation in 3D chromatin organization and regulation of gene expression. They introduce the concept of phase separation, the origins, biological implications and potential mechanisms. They also describe two previously proposed models: polymer-polymer (PPPS) and liquid-liquid phase separation (LLPS). Overall, the subject is very timely and of high interest for a broader and specialized audience. The text is comprehensive and crisp, with few exceptions discussed below, and the figure is very well-organized. Below are some suggestions that I believe will further improve the manuscript:

-The discussion regarding the two phase separation models is not very clear for a non-specialized reader. Why for example phase separation of CTCF and HP1 belongs to the PPPS and not the LLPS category?

I think a table summarizing critical distinctions between PPPS and LLPS will be tremendously beneficial for any reader and will elevate the value and impact of this review. The table could include: key distinctive points regarding the organizational principles, known structures that belong in each category with references, proteins and other factors previously implicated in each model, in vitro and in vivo assays that allow assessment of PPPS or LLPS, and biological processes affected by PPPS and/or LLPS.

-The structure of the review is a bit confusing too. The authors start by providing a background on chromatin organization, then they switch to phase separation and then to gene regulation prior to describing mechanisms of phase separation. I think they should commit to the role of PS in chromatin organization from compartmentalization and TADs all the way to enhancer-promoter interactions. Alternatively, they could start by introducing PS (models, mechanisms) and then discuss in different chapters the implications in chromosome architecture and gene regulation.

- The title of the last chapter is a bit misleading, since this section really focuses on the role of ATP in phase separation.

-Interesting points are raised in the outlook, however many of the raised questions are not directly related to chromatin regulation but instead focus on general concepts related to phase separation.

Minor comments:

-Some sentences in the text seem overcomplicated and difficult to follow. For example: Lines 85-88 and lines 103-105.

-Lines 27-29 may lead to misinterpretation, since they distinguish loop domains from TADs. Although, indeed these terms describe different principles of organization, the resulting topological features are highly overlapping. Therefore, I suggest replacing the "and" in line 28 with "and/or".

Author's Response to Decision Letter for (RSOB-19-0167.R0)

See Appendix A.

Decision letter (RSOB-19-0167.R1)

18-Sep-2019

Dear Professor Papantonis

We are pleased to inform you that your manuscript RSOB-19-0167.R1 entitled "Modes of phase separation affecting chromatin regulation" has been accepted by the Editor for publication in Open Biology. The reviewer(s) have recommended publication, but also suggest some minor revisions to your manuscript. Therefore, we invite you to respond to the reviewer(s)' comments and revise your manuscript.

Please submit the revised version of your manuscript within 14 days. If you do not think you will be able to meet this date please let us know immediately and we can extend this deadline for you.

- 1) A text file of the manuscript (doc, txt, rtf or tex), including the references, tables (including captions) and figure captions. Please remove any tracked changes from the text before submission. PDF files are not an accepted format for the "Main Document".
- 2) A separate electronic file of each figure (tiff, EPS or print-quality PDF preferred). The format should be produced directly from original creation package, or original software format. Please note that PowerPoint files are not accepted.
- 3) Electronic supplementary material: this should be contained in a separate file from the main text and meet our ESM criteria (see <http://royalsocietypublishing.org/instructions-authors#question5>). All supplementary materials accompanying an accepted article will be treated as in their final form. They will be published alongside the paper on the journal website and posted on the online figshare repository. Files on figshare will be made available approximately one week before the accompanying article so that the supplementary material can be attributed a unique DOI.

Online supplementary material will also carry the title and description provided during submission, so please ensure these are accurate and informative. Note that the Royal Society will not edit or typeset supplementary material and it will be hosted as provided. Please ensure that the supplementary material includes the paper details (authors, title, journal name, article DOI). Your article DOI will be 10.1098/rsob.2016[*last 4 digits of e.g. 10.1098/rsob.20160049*].

- 4) A media summary: a short non-technical summary (up to 100 words) of the key findings/importance of your manuscript. Please try to write in simple English, avoid jargon, explain the importance of the topic, outline the main implications and describe why this topic is newsworthy.

Images

Data-Sharing

It is a condition of publication that data supporting your paper are made available. Data should be made available either in the electronic supplementary material or through an appropriate repository. Details of how to access data should be included in your paper. Please see <http://royalsocietypublishing.org/site/authors/policy.xhtml#question6> for more details.

Data accessibility section

Sincerely,

The Open Biology Team
mailto:openbiology@royalsociety.org

Reviewer(s)' Comments to Author:

Author's Response to Decision Letter for (RSOB-19-0167.R1)

See Appendix B.

Decision letter (RSOB-19-0167.R2)

18-Sep-2019

Dear Professor Papantonis

We are pleased to inform you that your manuscript entitled "Modes of phase separation affecting chromatin regulation" has been accepted by the Editor for publication in Open Biology.

Sincerely,

The Open Biology Team
mailto:openbiology@royalsociety.org

Appendix A

University Medical Centre | Georg-August University of Göttingen
Institute of Pathology | Robert-Koch-Str. 40 | D-37075 Göttingen | Germany

Argyris Papantonis, Ph.D.
Professor for Translational Epigenetics

Phone: +49 551 39 65734

Fax: +49 551 39 8627

argyris.papantonis@med.uni-goettingen.de

Institute of Pathology,
University Medical Centre Göttingen,
Robert-Koch-Str. 40,
D-37075 Göttingen, Germany
<http://www.pathologie-umg.de>

To: the editorial office of *Open Biology*

Göttingen, 03.09.2019

Revision of invited review, RSOB-19-0167

Dear Prof. Glover,
Dear *OB* editorial team,

We would now like to resubmit our revised mini-review entitled "*Modes of phase separation affecting chromatin regulation*" to *Open Biology*. We would like to thank the reviewer for the careful comments, which we addressed in full (see point-by-point response on the next page). We hope that the manuscript is now ready for final acceptance and we look forward to seeing in press soon.

Sincere regards,

Prof. Dr. Argyris Papantonis

POINT-BY-POINT RESPONSE TO THE REVIEWER'S COMMENTS

Comments to the Author(s)

In this review, Palikyras and Papantonis, discuss recent findings that support the role of phase separation in 3D chromatin organization and regulation of gene expression. They introduce the concept of phase separation, the origins, biological implications and potential mechanisms. They also describe two previously proposed models: polymer-polymer (PPPS) and liquid-liquid phase separation (LLPS). Overall, the subject is very timely and of high interest for a broader and specialized audience. The text is comprehensive and crisp, with few exceptions discussed below, and the figure is very well-organized. Below are some suggestions that I believe will further improve the manuscript:

We would like to thank the reviewer for the encouraging words, as well as for all the suggestions below.

- The discussion regarding the two phase separation models is not very clear for a non-specialized reader. Why for example phase separation of CTCF and HP1 belongs to the PPPS and not the LLPS category? I think a table summarizing critical distinctions between PPPS and LLPS will be tremendously beneficial for any reader and will elevate the value and impact of this review. The table could include: key distinctive points regarding the organizational principles, known structures that belong in each category with references, proteins and other factors previously implicated in each model, in vitro and in vivo assays that allow assessment of PPPS or LLPS, and biological processes affected by PPPS and/or LLPS.

The reviewer is right, this point can come across as blurry. The suggestion for the addition of a Table is a very good one, and we now provide all the information suggested in new **Table 1**.

- The structure of the review is a bit confusing too. The authors start by providing a background on chromatin organization, then they switch to phase separation and then to gene regulation prior to describing mechanisms of phase separation. I think they should commit to the role of PS in chromatin organization from compartmentalization and TADs all the way to enhancer-promoter interactions. Alternatively, they could start by introducing PS (models, mechanisms) and then discuss in different chapters the implications in chromosome architecture and gene regulation.

We appreciate the validity of the suggestion, and have now restructured the order of the section in this new version of the manuscript.

- The title of the last chapter is a bit misleading, since this section really focuses on the role of ATP in phase separation.

This last part discusses the idea of ATP in PS, but also how modifications like PARylation in IDRs can take up similar roles. Thus, we still think that the title needs to refer to “*some mechanisms*” (rather than only to one), although we acknowledge that there is not much elaboration on the non-ATP-related part.

- Interesting points are raised in the outlook, however many of the raised questions are not directly related to chromatin regulation but instead focus on general concepts related to phase separation.

We find that by addressing these points on phase separation, novel insights for chromatin regulation will also be illuminated. Hence, we now clarify this in a new sentence, but have left the rest of this outlook section essentially unchanged.

Minor comments:

- Some sentences in the text seem overcomplicated and difficult to follow. For example: Lines 85-88 and lines 103-105.

We have now revisited these sentences to simplify them.

- Lines 27-29 may lead to misinterpretation, since they distinguish loop domains from TADs. Although, indeed these terms describe different principles of organization, the resulting topological features are highly overlapping. Therefore, I suggest replacing the “and” in line 28 with “and/or”.

This is a fair point, since TADs and loops domains might indeed converge; “and/or” is now used in the text.

Appendix B

University Medical Centre | Georg-August University of Göttingen
Institute of Pathology | Robert-Koch-Str. 40 | D-37075 Göttingen | Germany

Argyris Papantonis, Ph.D.
Professor for Translational Epigenetics

Phone: +49 551 39 65734

Fax: +49 551 39 8627

argyris.papantonis@med.uni-goettingen.de

Institute of Pathology,
University Medical Centre Göttingen,
Robert-Koch-Str. 40,
D-37075 Göttingen, Germany
<http://www.pathologie-umg.de>

To: the editorial office of *Open Biology*

Göttingen, 03.09.2019

Revision of invited review, RSOB-19-0167

Dear Prof. Glover,
Dear *OB* editorial team,

We would now like to resubmit our revised mini-review entitled "*Modes of phase separation affecting chromatin regulation*" to Open Biology. We would like to thank the reviewer for the careful comments, which we addressed in full (see point-by-point response on the next page). We hope that the manuscript is now ready for final acceptance and we look forward to seeing in press soon.

Sincere regards,

Prof. Dr. Argyris Papantonis

POINT-BY-POINT RESPONSE TO THE REVIEWER'S COMMENTS

Comments to the Author(s)

In this review, Palikyras and Papantonis, discuss recent findings that support the role of phase separation in 3D chromatin organization and regulation of gene expression. They introduce the concept of phase separation, the origins, biological implications and potential mechanisms. They also describe two previously proposed models: polymer-polymer (PPPS) and liquid-liquid phase separation (LLPS). Overall, the subject is very timely and of high interest for a broader and specialized audience. The text is comprehensive and crisp, with few exceptions discussed below, and the figure is very well-organized. Below are some suggestions that I believe will further improve the manuscript:

We would like to thank the reviewer for the encouraging words, as well as for all the suggestions below.

- The discussion regarding the two phase separation models is not very clear for a non-specialized reader. Why for example phase separation of CTCF and HP1 belongs to the PPPS and not the LLPS category? I think a table summarizing critical distinctions between PPPS and LLPS will be tremendously beneficial for any reader and will elevate the value and impact of this review. The table could include: key distinctive points regarding the organizational principles, known structures that belong in each category with references, proteins and other factors previously implicated in each model, in vitro and in vivo assays that allow assessment of PPPS or LLPS, and biological processes affected by PPPS and/or LLPS.

The reviewer is right, this point can come across as blurry. The suggestion for the addition of a Table is a very good one, and we now provide all the information suggested in new **Table 1**.

- The structure of the review is a bit confusing too. The authors start by providing a background on chromatin organization, then they switch to phase separation and then to gene regulation prior to describing mechanisms of phase separation. I think they should commit to the role of PS in chromatin organization from compartmentalization and TADs all the way to enhancer-promoter interactions. Alternatively, they could start by introducing PS (models, mechanisms) and then discuss in different chapters the implications in chromosome architecture and gene regulation.

We appreciate the validity of the suggestion, and have now restructured the order of the section in this new version of the manuscript.

- The title of the last chapter is a bit misleading, since this section really focuses on the role of ATP in phase separation.

This last part discusses the idea of ATP in PS, but also how modifications like PARylation in IDRs can take up similar roles. Thus, we still think that the title needs to refer to “*some mechanisms*” (rather than only to one), although we acknowledge that there is not much elaboration on the non-ATP-related part.

- Interesting points are raised in the outlook, however many of the raised questions are not directly related to chromatin regulation but instead focus on general concepts related to phase separation.

We find that by addressing these points on phase separation, novel insights for chromatin regulation will also be illuminated. Hence, we now clarify this in a new sentence, but have left the rest of this outlook section essentially unchanged.

Minor comments:

- Some sentences in the text seem overcomplicated and difficult to follow. For example: Lines 85-88 and lines 103-105.

We have now revisited these sentences to simplify them.

- Lines 27-29 may lead to misinterpretation, since they distinguish loop domains from TADs. Although, indeed these terms describe different principles of organization, the resulting topological features are highly overlapping. Therefore, I suggest replacing the “and” in line 28 with “and/or”.

This is a fair point, since TADs and loops domains might indeed converge; “and/or” is now used in the text.